# Trip duration drives shift in travel network structure with implications for the predictability of spatial disease spread

**John R. Giles**[1]\*, **Derek A. T. Cummings**[2], **Bryan T. Grenfell**[3], **Andrew J. Tatem**[4], **Elisabeth zu Erbach-Schoenberg**[4], **CJE Metcalf**[3], **Amy Wesolowski**[1]

**1** Department of Epidemiology, Johns Hopkins Bloomberg School of Public Health, Baltimore, Maryland, United States of America, **2** Department of Biology and the Emerging Pathogens Institute, University of Florida, Gainesville, Florida, United States of America, **3** Department of Ecology and Evolutionary Biology and the Princeton School of Public and International Affairs, Princeton University, Princeton, New Jersey, United States of America, **4** WorldPop, School of Geography and Environmental Science, University of Southampton, Southampton, United Kingdom

\* giles@jhu.edu

**Data Availability Statement:** Due to the data-sharing agreement with the mobile phone provider, the authors may not distribute the data used here.

## Abstract

Human travel is one of the primary drivers of infectious disease spread. Models of travel are often used that assume the amount of travel to a specific destination decreases as cost of travel increases with higher travel volumes to more populated destinations. Trip duration, the length of time spent in a destination, can also impact travel patterns. We investigated the spatial patterns of travel conditioned on trip duration and find distinct differences between short and long duration trips. In short-trip duration travel networks, trips are skewed towards urban destinations, compared with long-trip duration networks where travel is more evenly spread among locations. Using gravity models to inform connectivity patterns in simulations of disease transmission, we show that pathogens with shorter generation times exhibit initial patterns of spatial propagation that are more predictable among urban locations. Further, pathogens with a longer generation time have more diffusive patterns of spatial spread reflecting more unpredictable disease dynamics.

## Author summary

During an epidemic of an infectious pathogen, cases of disease can be imported to new locations when people travel. The amount of time that an infected person spends in a destination (trip duration) determines how likely they are to infect others while travelling. In this study, we analyzed travel data and found specific spatial patterns in trip duration, where short-duration trips are more common between urban destinations and long-duration trips are evenly spread out among locations. To show how this spatial pattern impacts the spread of infectious diseases, we used data-driven models and simulations to show that pathogens with shorter generation times have patterns of spatial spread that are more predictable among urban locations. However, pathogens with longer generation times

These data can be requested directly from MTC mobile (https://www.mtc.com.na/contact/contactdetails). All other relevant data are within the manuscript and its Supporting Information files.

**Funding:** JRG and AW are supported by the National Library Of Medicine of the National Institutes of Health under Award Number DP2LM013102. The content is solely the responsibility of the authors and does not necessarily represent the official views of the National Institutes of Health. APW is also funded by a Career Award at the Scientific Interface by the Burroughs Wellcome Fund. The funders had no role in study design, data collection and analysis, decision to publish, or preparation of the manuscript.

**Competing interests:** The authors have declared that no competing interests exist.

tend to spread along the long-duration travel networks that are more evenly distributed among locations giving them more unpredictable disease dynamics.

## Introduction

During an infectious disease outbreak, anticipating where a pathogen will spread is an important part of planning an effective response [1,2]. The initial stages of an outbreak are typically characterized by spatial dynamics through the population from an introduction event, such as the 2014 Ebola outbreak in West Africa [3] and the COVID-19 pandemic [4] which started at the end of 2019. These outbreaks provide clear examples of how quickly a pathogen can spread through a highly susceptible population. The speed of spatial propagation depends not only on population susceptibility and intrinsic biological properties of the pathogen, but also on connectivity patterns driven by human travel [2,5]. Mathematical models have been used to predict this type of spatial dynamics, which has been crucial for informing deployment of interventions [1,6] for a range of pathogens including Ebola [7–9], influenza [10–12], SARS-CoV-1 [13–15], and the ongoing COVID-19 pandemic caused by SARS-CoV-2 [16–18]. However, mathematical models of disease spread often exhibit complex dynamics [19–21] that have a limited prediction horizon [22] due to the inherent complexities of both disease transmission and human behavior. Therefore, recent methods have relied more heavily on detailed information about human mobility patterns [23–25] to inform the spatial dynamics of disease spread.

Spatial connectivity in such models frequently relies on data and/or models of human mobility [2,26]. Ideally, mobility is derived from travel data such as travel surveys [27,28], call data records [29–31], air traffic data [8,32,33] or App data [34,35], but when these data are absent or incomplete, mechanistic models of human movement are often used [36–39]. These data and models are typically used to provide spatial structure in a metapopulation framework [40] of disease transmission, where subpopulations are treated as discrete units that are connected by travel volume (raw number or relative magnitude of trips among locations per unit time). However, the extent to which human mobility patterns can explain spatial disease dynamics depends on whether or not travel volume scales directly with the rate of transmission among subpopulations. While a direct-scaling scenario conveniently assumes a constant probability of onward transmission for all trips, the manner in which mobility scales with transmission depends on the specifics of epidemiological-relevant movement among locations. In particular, the number of potentially infectious contacts that occur between individuals [41–43]—which will depend largely on the duration of a trip [44,45]—is a key factor that determines the likelihood of onward transmission and the epidemiological-relevance of observed travel volume.

One way in which travel volume may not scale directly to transmission among subpopulations is if the amount of time that travelers spend in a destination (trip duration) varies spatially. Previous work has explored scaling transmission based on trip duration, typically as some return rate. Return rate methods use a diffusion rate of travel that is then reduced based on the typical interval of time spent in a destination, this effectively scales down transmission per route based on assumptions about travel behavior [36,46–49]. These methods have been applied where all routes are scaled down uniformly [36] or according to the origin [46], or in the case of Poletto et al. [48,49] where return rate was determined functionally according to the degree distribution of the destination. It follows that the duration of trips may depend on the origin and destination location which would suggest that integrating trip duration data

would not uniformly change spatial transmission patterns [50,51]. For example, studies of urban travel have found that trip duration is associated with the attractiveness of a destination (e.g. work, residence, recreation) [52–55] and that differences among locations form discrete spatial regions with unique patterns of trip duration [56–58]. Therefore, taking trip duration into account and the spatial heterogeneities that influence the length of stay when individuals travel may help to refine spatial connectivity in epidemiological models so that patterns of spatial spread are more representative of actual routes of transmission allowing more informed prediction spatial disease spread.

Here, we use mobile phone data from Namibia to examine how networks of human travel change based on the duration of trips. We find that short trip duration networks exhibit heterogeneous patterns of connectivity that shift to homogeneous patterns as trip duration increases. We show that spatial patterns in trip duration may be driven by differential aspects of travel behavior, where short duration trips are strongly influenced by the cost of travel (measured by travel distance and population size), but the cost of travel is less important for longer trips. To show how these travel patterns influence disease dynamics, we use gravity models fitted to the mobile phone data to develop novel models of spatial connectivity and simulate spatial disease spread for a range of pathogens and points of introduction. We demonstrate that gravity model parameters can be adjusted to account for shifts in network heterogeneity due to trip duration and show that the more diffuse connectivity for longer-duration travel has important implications for the predictability of disease spread.

## Results

We used an anonymized data set of call data records (CDRs) collected in Namibia from October 2010 to April 2014 to estimate mobility patterns. Previous studies have used versions of these CDRs and provide detailed descriptions of the data [51,59,60]. Briefly, mobile phone usage data from 2.2 million unique subscribers was used to estimate travel patterns based on the estimated daily location of each subscriber, where the daily location is determined by recording the most frequently used mobile phone tower for each subscriber each day [29,61]. A trip was recorded when the location of a subscriber changed from location A to location B on subsequent days, otherwise the subscriber was recorded as staying in location A. For each trip, the date of travel and the number of days the subscriber remained in the destination location (trip duration) was recorded. The locations of the mobile phone towers were then aggregated to the 105 administrative level 2 regions (districts) in Namibia (Fig 1). For each day, these data provide an estimate for the number of trips made among the 105 districts and the duration of each trip, which include 259.2 million trips during the 40 months of data collection.

To explore the impact of trip duration on connectivity patterns, we analyzed patterns of travel by assessing the spatial distribution of trips based on different trip durations. Short-duration trips (e.g. trips lasting between 1–3 days) were characterized by a high proportion of trips to nearby destinations with larger populations. There were fewer long-trip duration trips (30–60 days) which were more evenly distributed across all locations without a clear skew to nearby destinations with larger populations (see Fig 2). These patterns change incrementally when assessed over a wider range of 20 subnetworks defined by trip duration reflecting finer time intervals that are analogous to the generation times of several infectious pathogens (see Table 1). For each subnetwork, we treated each location as a node and calculated the statistical distributions of two network centrality measures: node strength (node degree weighted by total number of trips) and node closeness (total weighted distance from all nodes) using Barrat's method [62], see Methods for more detailed description of these metrics. We found that

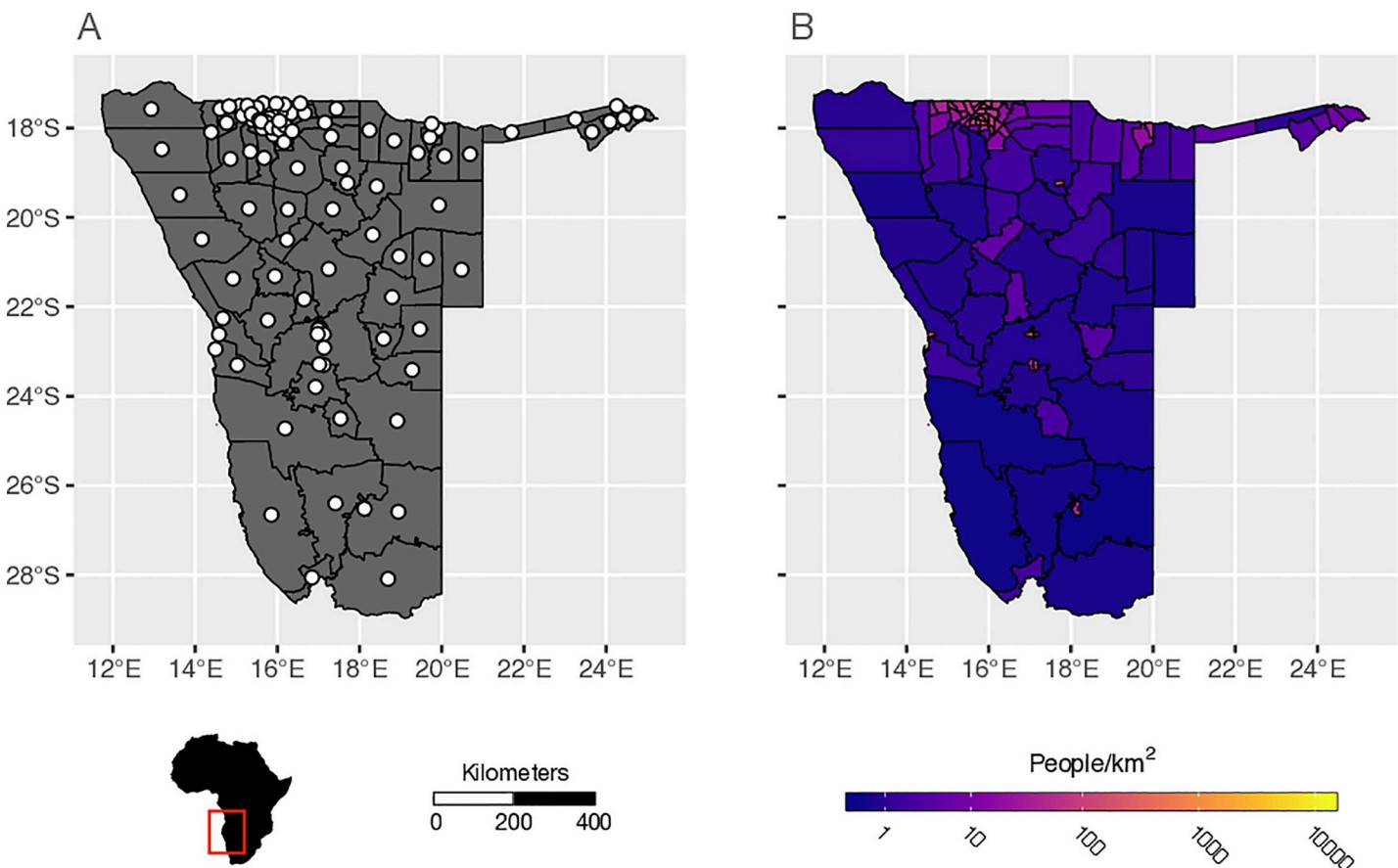

**Fig 1. The Administrative districts and population distribution in Namibia.** A) The locations of the 105 districts (administrative level 2) in Namibia with district centroids of each shown with white circles. B) The population density of each district in calculated as the number of people per square kilometer from the 2010 WorldPop Project estimates of the total number of people per 100m grid cell within each district (www.worldpop.org). District-level shapefiles were acquired from DIVA-GIS (www.diva-gis.org) and are reused here under the Creative Commons Attribution License.

for short-trip duration networks, the average expected values for both node strength and closeness values were high with an overall distribution that was flatter, which implies that there are particular nodes (high population density urban locations) that are more highly connected (S3 and S4 Figs). In contrast, the long-trip networks, had distributions of node strength and closeness that were lower and more uniform, which implies that these networks are less connected, and travel is more evenly distributed across all locations (see Figs 3A, 3B and S2).

To more directly quantify the manner that spatial connectivity shifts for longer trip duration networks, we used a multivariate changepoint algorithm to identify statistically significant changes in network heterogeneity as metrics $\eta_{\text{strength}}$ and $\eta_{\text{close}}$ (see Methods for more detailed description of these metrics), which indicate durations of travel at which network topology shifts significantly from heterogeneous to homogenous. We identified statistically significant shifts in network topology at 5 and 60 days (*p*-value = 1e-03 and 3e-04 respectively), which divides the duration-restricted subnetworks into three nominal classes: heterogeneous (1–5 days), intermediate (6–60 days), and homogeneous (>60 days; see Fig 3C). Overall, these results suggest that trips with a duration of less than 5 days display connectivity that is highly clustered among densely populated areas (Class 1 in Fig 3C), but trips longer than 60 days have more homogenously distributed trip volume among all locations (Class 3 in Fig 3C). In between these extremes is an intermediate class where trips are more evenly distributed

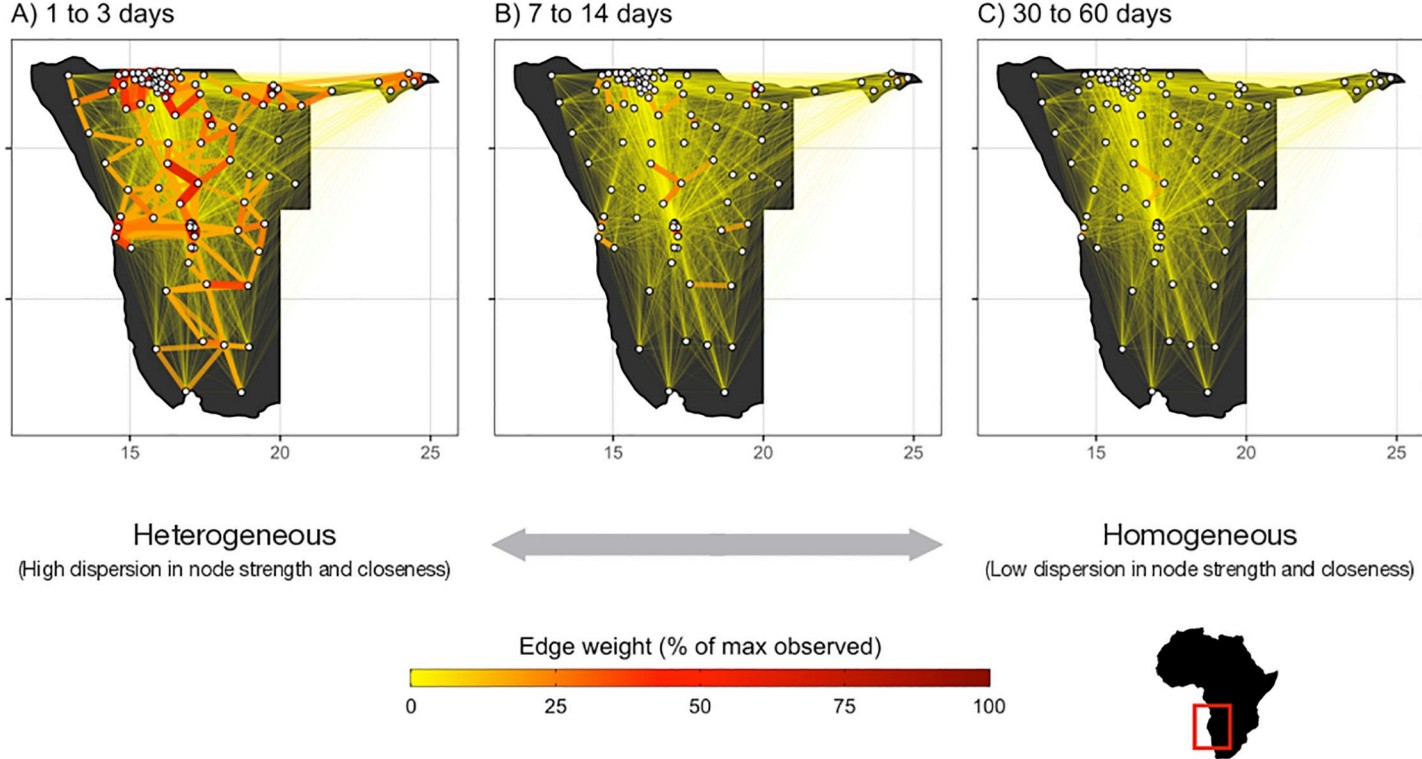

**Fig 2. Travel network topology shifts from heterogeneous to homogeneous as trip duration increases.** Maps of Namibia with travel volumes between districts that fall within three broad intervals of trip duration: A) 1–3 days, B) 7–14 days, and C) 30–60 days. For comparison, connectivity is defined as relative edge weight from 0 to 100%, which is calculated by scaling trip volume along the edges in each sub-network by the overall maximum trip volume observed in the full travel network. District centroids (nodes of the network) are indicated by the white circles. Shapefiles were acquired from DIVA-GIS (www.diva-gis.org) and are reused here under the Creative Commons Attribution License.

among all locations, but there is still some clustering among more densely populated locations (Class 2 in Fig 3C).

To understand factors driving differences in connectivity, we fit log-linear regression models to each of the 105 origin locations in the Namibia mobile phone data. For each location, the number of trips observed from the origin to each of its destinations was estimated using variables describing characteristics of the origin and destination: distance, total population size, population density (people/km$^2$), and total administrative area (km$^2$). These models were applied to the trip counts within each duration-restricted subnetwork to understand how these covariates drive trip volume across progressively longer intervals of trip duration. Overall, we tested 13 candidate log-linear regression models with relevant combinations of these variables and found that a model containing i) distance between origin and destination, and ii) the destination's population size was the most parsimonious best fitting model (see S9 Fig). With this model specification, we estimated the effect size (fitted slope) and found that distance had a negative effect consistent with trip counts that decay as distance increases and destination population size had a net positive effect (i.e. more trips attracted to densely populated places). Since these covariates have rather different scales, we standardized them based on their standard deviation to allow for comparison and we found that on average, distance had an effect size that was on average 6.7 (0.2–17.3 95% CI) times larger than destination population size (mean effect size across all locations and trip durations was -1.02 and 0.6 respectively), indicating that in largely rural Namibia, distance remains the primary driver of trip volume

 

**Table 1. Table of trip duration intervals used, with number observations and total trip counts.**

| Duration interval (days) | | | | Number routes observed | Number total trips (10,000s) |
|---|---|---|---|---|---|
| Start | Stop | Model | Description | | |
| 0 | Inf | 0 | All durations | 10843 | 25926 |
| 0 | 1 | 1 | 0 to 1 days | 10508 | 12083 |
| 1 | 2 | 2 | 1 to 2 days | 10629 | 16324 |
| 2 | 3 | 3 | 2 to 3 days | 10391 | 6376 |
| 3 | 5 | 4 | 3 to 5 days | 10379 | 4455 |
| 5 | 7 | 5 | 5 to 7 days | 10173 | 2169 |
| 7 | 10 | 6 | 7 to 10 days | 10127 | 1444 |
| 10 | 14 | 7 | 10 to 14 days | 9977 | 1076 |
| 14 | 21 | 8 | 2 to 3 weeks | 10013 | 863 |
| 21 | 28 | 9 | 3 to 4 weeks | 9740 | 475 |
| 30 | 60 | 10 | 1 to 2 months | 10044 | 617 |
| 60 | 90 | 11 | 2 to 3 months | 9274 | 203 |
| 90 | 120 | 12 | 3 to 4 months | 8505 | 98 |
| 120 | 150 | 13 | 4 to 5 months | 7594 | 49 |
| 150 | 180 | 14 | 5 to 6 months | 6892 | 29 |
| 180 | 210 | 15 | 6 to 7 months | 6173 | 19 |
| 210 | 240 | 16 | 7 to 8 months | 5706 | 13 |
| 240 | 270 | 17 | 8 to 9 months | 4943 | 9 |
| 270 | 300 | 18 | 9 to 10 months | 4379 | 6 |
| 300 | 330 | 19 | 10 to 11 months | 4052 | 5 |
| 330 | 360 | 20 | 11 to 12 months | 3819 | 4 |

regardless of trip duration. We also found that effect sizes for both covariates were strongest for short duration trips ($\approx 5$ days or less), but these effects are reduced to near-zero for longer trip durations (Fig 4C and 4D), and for some origin districts the effect of destination population was reversed for longer trips ($> 60$ days), which suggests that districts with lower population density may even attract more long duration trips (Fig 4B and 4D).

We assessed the ability of a simple gravity model to capture the changes in spatial connectivity over each of the 20 duration-restricted subnetworks shown in Table 1. When we compared gravity models fitted to each subnetwork to that of a full travel network with all trips, we found that the duration-restricted models are similar to the full model containing all trip durations up to 7–10 days duration (subnetwork 6), after which both the number of trips and connectivity values estimated by the gravity models begin to decrease, becoming essentially uniformly distributed after 5–6 months (subnetwork 14; see Fig 5A). The changes in spatial connectivity in these models result from incremental decreases in the distance parameter $\gamma$ and destination population size parameter $\omega_2$, where both were essentially zero after 5–6 months duration (Fig 5B and 5C), indicating that distance and population size of the destination have little influence on the spatial distribution of these longer duration trips. Further, model fit decreased as trip duration increased, with $R^2$ values showing good model fit up to 1–2 months duration (subnetwork 10; $R^2 = 0.51$–$0.55$), but goodness of fit decreased for longer durations (see S6 Fig). However, since the observed number of routes and trips decrease for trips of longer durations (Table 1 and S1 Fig), this could spuriously cause the observed decrease in model fit (S6 Fig). To ensure that connectivity patterns estimated by these models were not just due to the reduced number of observed routes or trip counts, we artificially down-sampled the full data according to the observed routes and sample sizes in Table 1. We

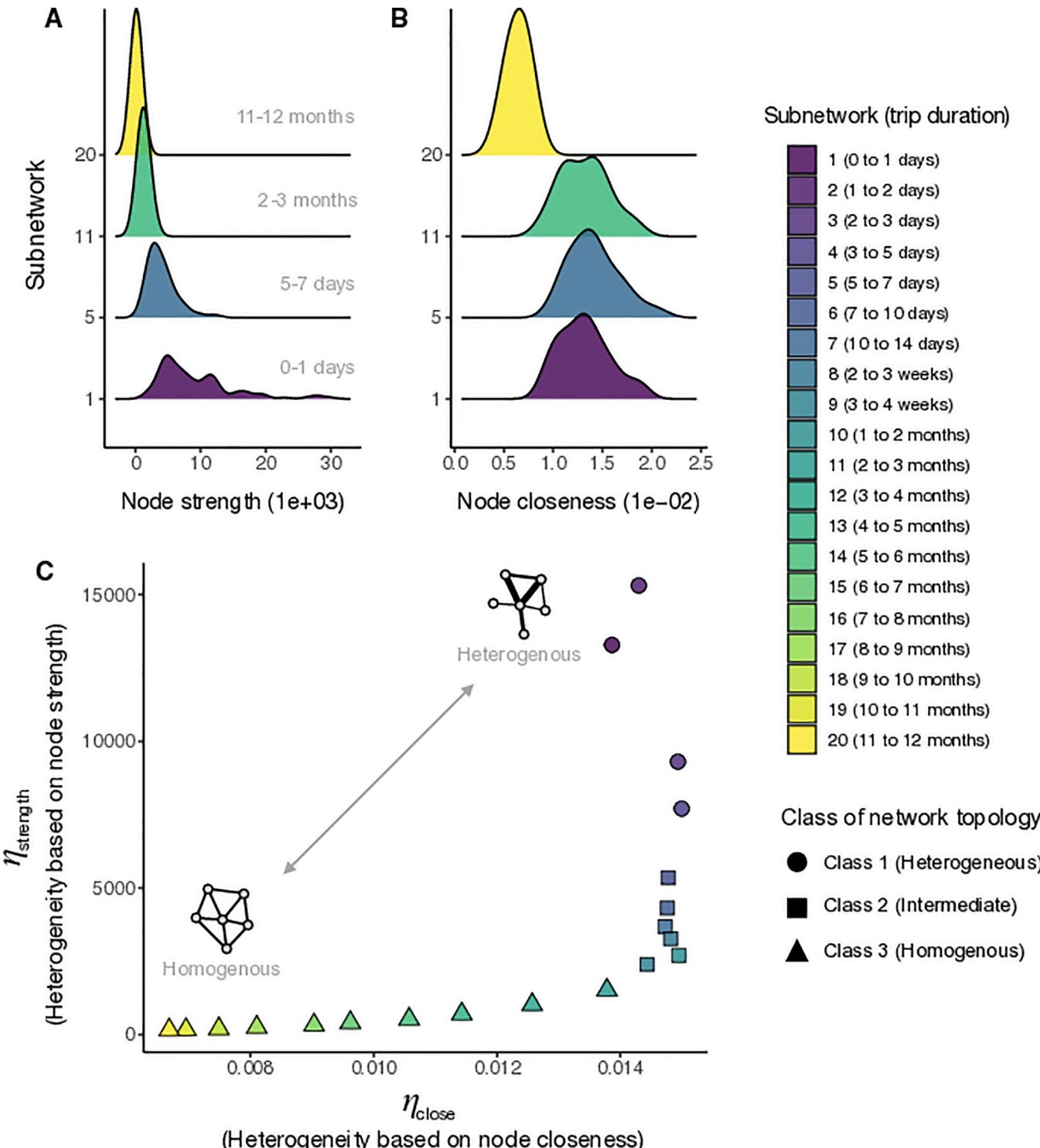

**Fig 3. Distribution of node strength and node centrality for models with different trip duration intervals and changepoint analysis showing which trip durations constitute shifts in network topology.** The empirical distributions of A) node strength (weighted degree per node) and B) node closeness (total weighted distance from all nodes) are plotted for four duration-restricted subnetworks which include trips of 1) 0–1 days, 5) 5–7 days, 11) 2–3 months, and 20) 11–12 months. C) The joint distribution of $\eta_{strength}$ and $\eta_{close}$ which measures the amount of network heterogeneity and structure based on the distributions of node strength and node closeness respectively. Each point represents one of the duration-restricted subnetworks and is colored according the duration interval shown in the color key to the right. The multivariate changepoint algorithm identified two significant shifts in network topology based on the joint distribution

of $\eta_{strength}$ and $\eta_{close}$ that are placed at 5 and 60 days trip duration. The three nominal classes delineated by these thresholds are indicated by a circle for the heterogeneous class (1–5 days), a triangle for the homogeneous class (>60 days), and a square for intermediate class (6–60 days).

found that the connectivity values estimated by gravity models fitted to artificially down-sampled subnetworks were largely the same as the full model (S7 and S8 Figs), which indicates that the difference in spatial patterns among the actual subnetworks are robust to the smaller numbers of observations or trip counts.

## Predictability of spatial spread

To explore how shifts in travel network connectivity impact predictability of spatial disease spread, we first modeled the spatial connectivity of disease spread for a range of pathogen life histories (S1 Table) and then estimated the spatial force of infection conditioned on the generation time of each pathogen (see Methods). Starting with an initially infected location, we then assessed how reliably we can identify the next location where the disease would spread for each successive generation using an index of spatial predictability ($\phi$). Spatial predictability $\phi$ is based on the principle of maximum entropy and provides an easily interpretable index between 0 and 1 that quantifies how the probability of disease importation is distributed among destinations—see Methods for detailed description.

When we analyzed the spatial simulations for each pathogen, we found that, given the same starting location (capital district of Windhoek), pathogens with shorter generation times and low $R_0$ had the most predictable spatial dynamics in the initial stages of an outbreak (Fig 6A). In comparison, pathogens with longer generation times had less predictable spatial dynamics. However, this varied by pathogen characteristics where a high $R_0$ can compensate for a short generation time making initial dynamics less predictable because these pathogens are able to spread to multiple destinations very quickly. Beyond initial dynamics, predictability remained largely constant with successive generations for most pathogens (Fig 6B). The mean value of spatial predictability for simulations of malaria showed a downward trend in predictability with subsequent generations, however transmission in these simulations tended to fade out after 8–10 generations and had wide confidence intervals around the mean. Further, in simulations of Ebola and SARS-CoV-2, spatial predictability dips (likely when highest rates of spatial spread are occurring mid-epidemic) but then return to initial levels. But in the case of measles, with a high $R_0$, the reduction in predictability is less pronounced because it has a faster rate of transmission and associated spatial spread. This suggests that, while the impact of travel network heterogeneity has a rather straightforward effect on spatial predictability when considering generation time, local dynamics driving the growth rate of the infected population in the index location (i.e. $R_0$, proportion susceptible) may influence spatial predictability in more complex ways.

These patterns of spatial predictability were also dependent on the chosen starting location for the initial infected case in each simulation. For all pathogens, we found that predictability was highest for introduction into densely populated districts (Fig 7). Interestingly, we also found that the highest value of initial spatial predictability was $\phi = 0.56$ (a SARS-CoV-2 simulation with an introduction event near the capital district) meaning that, given the connectivity patterns observed in Namibia, in the best-case scenario spatial spread appears to be marginally more predictable than unpredictable. We investigated this theoretically and found that a value of $\phi = 0.56$ is functionally equivalent to the expectation that transmission is likely to occur from a given origin to 7 out of the 104 potential destinations (S11A Fig). In instances where

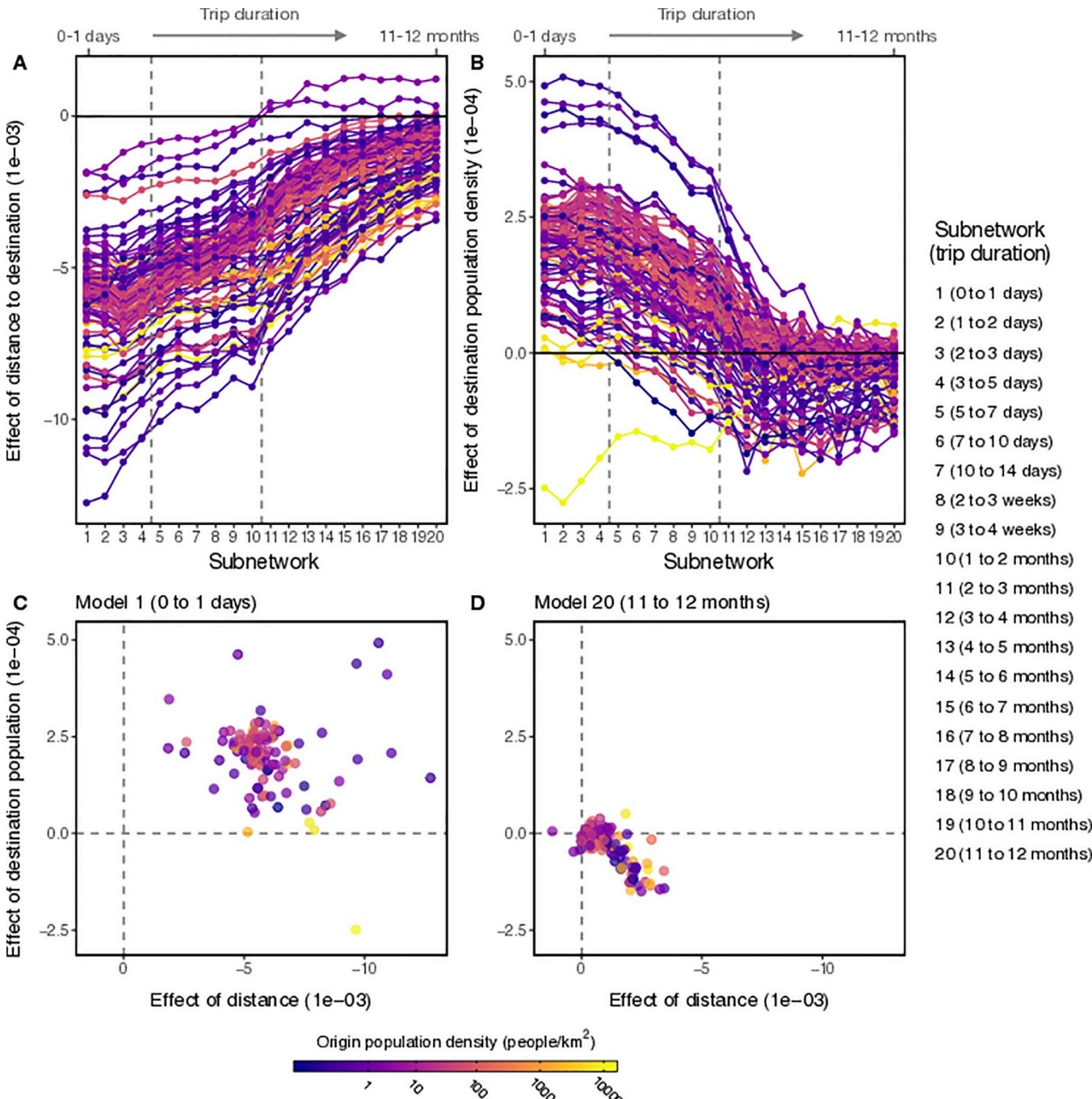

**Fig 4. The drivers of trip volume across varying trip durations.** The effect of distance to destination A) and destination population size B) on trip volume plotted across the 20 duration-restricted subnetworks. Each colored line represents one of the 105 districts with the population density of the origin district is indicated by the color bar. Dashed vertical lines indicate the network topology thresholds identified by the changepoint analysis. Scatterplots showing the joint distribution of effect sizes for distance (x-axis) and destination population (y-axis) for C) the model with the shortest duration (Model 1, 0–1 days) and D) the model with the longest duration (Model 20, 11–12 months). Comparison of C and D show that the effect size of both covariates is reduced to near-zero effect for longer trip durations.

the origin has more than one travel destination, which is common in highly connected travel networks, perfect certainty ($\phi = 1$) is exceedingly unlikely (S11B Fig). Although these results suggest that in most instances, we can expect spatial spread to be generally more unpredictable

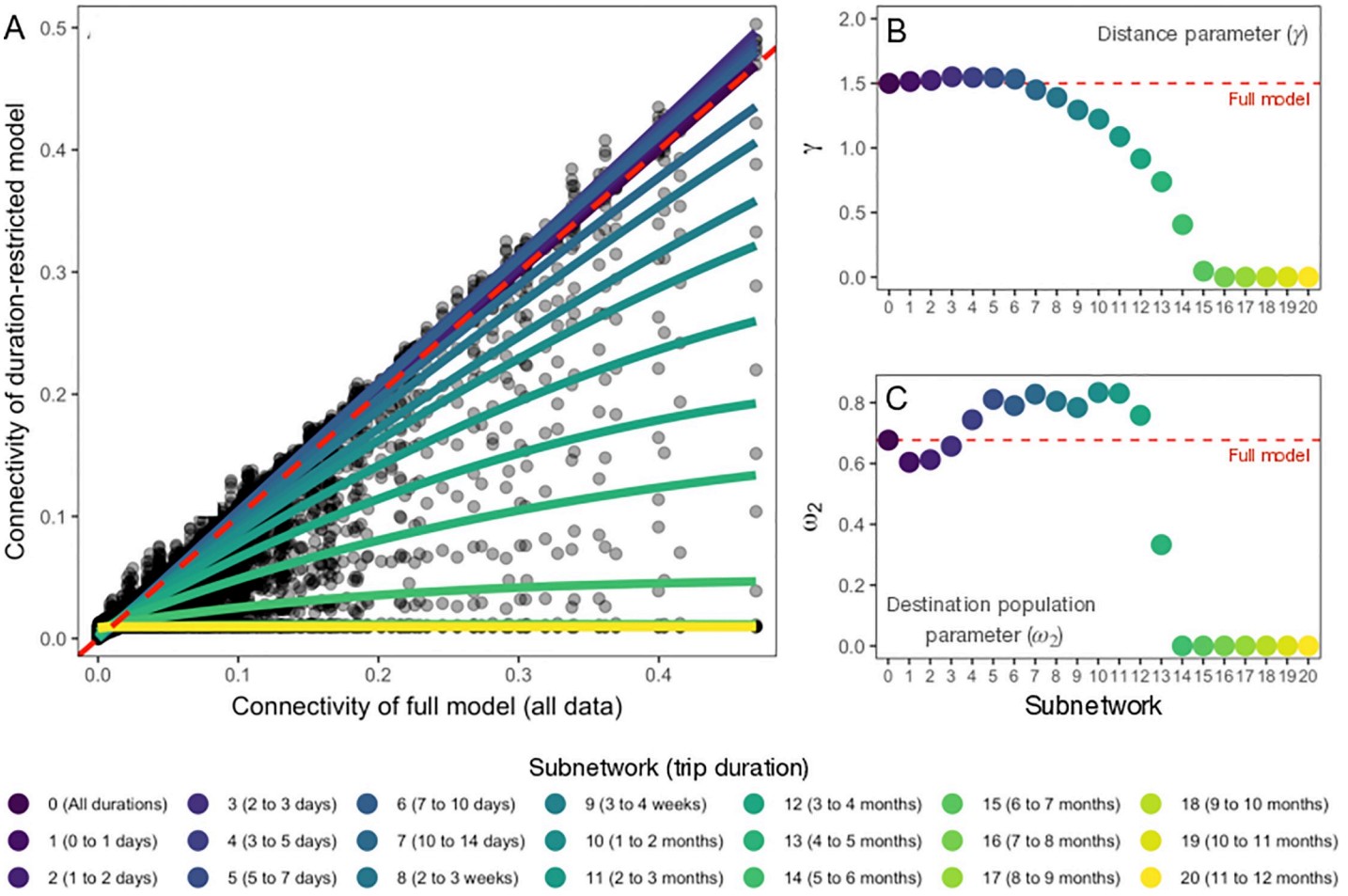

**Fig 5. Change in connectivity and gravity model parameters fitted to travel data with increasing duration intervals compared to full model.** A) The distribution of connectivity values for duration-restricted models (y-axis) in comparison to the full model that includes all data (x-axis). The smoothed lines indicate the change in connectivity for duration-restricted models with larger duration intervals showing a more evenly distributed pattern across all locations compared to the null model. The dashed red line indicates connectivity values that are equal to the full model. In B) and C), the change in fitted gravity model parameters (distance parameter $\gamma$ and destination population parameter $\omega_2$ respectively) for increasing trip duration intervals. The color gradient indicates the duration interval of each model and the dashed red line shows the fitted parameter value for the full model, which includes all trip durations.

than it is predictable, but that predictability remains marginally higher for pathogens introduced into urban areas with potential decreased predictability during peak epidemic generations.

## Discussion

There is often limited data that can provide both the number and duration of trips to model spatial connectivity in disease transmission models [31]. Previous work has shown that trip duration is subject to economic constraints due to the cost of travel [52,63] and these constraints often depend on characteristics of the destination [52–55]. The impact of these types of travel decisions on spatial connectivity and disease transmission has been explored theoretically using metapopulation models, but few studies have quantified this effect empirically with data that span larger spatial and temporal scales [46–49,51]. We used mobile phone data from Namibia to show that a travel network can be broken down into subnetworks based on trip duration. When we account for travel of different durations in these subnetworks, we find

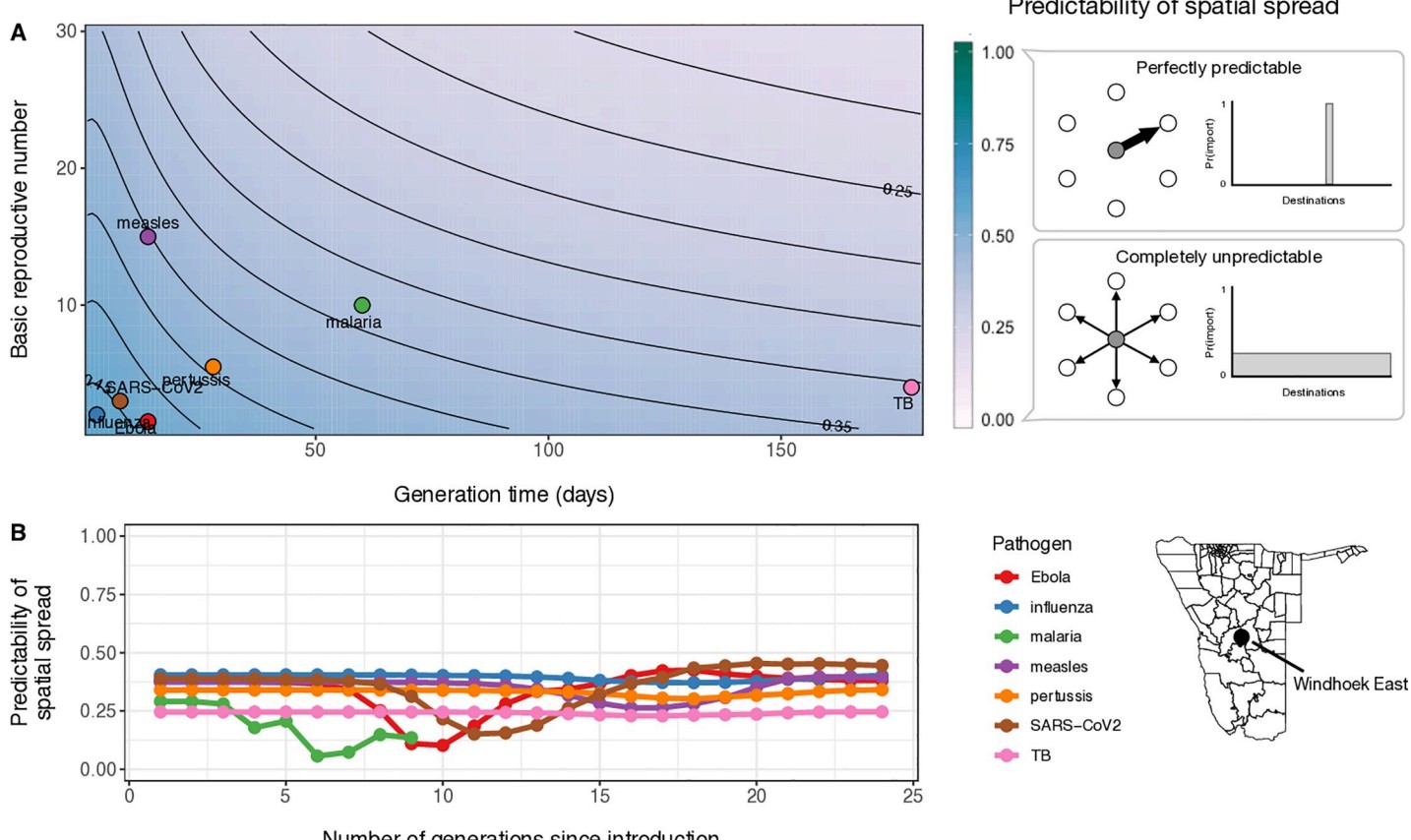

**Fig 6. Predictability of spatial spread for a range of $R_0$ and generation time values and the change in spatial predictability over time for 7 pathogens.** A) A heatmap with contour lines showing the values of initial spatial predictability ($\phi$) calculated for hypothetical combinations of $R_0$ and generation time (days). Example pathogens are indicated by colored circles: influenza, SARS-CoV-2, measles, Ebola, pertussis, *P. falciparum* malaria, tuberculosis. The level of spatial predictability is shown in the color bar to the right with schematic representations for scenarios where patterns of spatial spread are perfectly predictable ($\phi = 1$) or completely unpredictable ($\phi = 0$). B) The change in these initial values of spatial predictability over successive generations for each of the 7 example pathogens. Colored lines represent the mean value of spatial predictability calculated over 1000 replicates of the stochastic TSIR model (see Methods). In both analyses, the capital district of Namibia, Windhoek East, was used as the introduction district.

different spatial patterns in connectivity, where short-trip networks can be characterized by high node degree strength and node closeness with connectivity that is concentrated among densely populated areas. Conversely, long-trip networks can be characterized by lower node degree strength and node closeness, suggesting a spatial pattern of connectivity that is more evenly dispersed across all nodes, including those in low density locations. These spatial differences in network topology may be driven by travel decisions stemming from the cost of travel, where the different classes of subnetworks may emerge from constraints on trips depending on the duration [64]. Accordingly, we find that short trips are more impacted by distance thus these trips concentrate around densely populated locations and show more heterogeneous structure like scale-free networks. But longer duration trips are less constrained by the cost of travel and therefore show more homogeneous network topology.

Spatial heterogeneity in trip duration impacts infectious disease spread differentially through interaction with generation times of each pathogen. Keeling and Rohani [47] provide a theoretical precedent for the interdependence between pathogen life history and patterns of spatial connectivity. They show that when trip duration is shorter than the infectious period of the pathogen, the effective level of spatial coupling among locations is reduced. Similarly, we

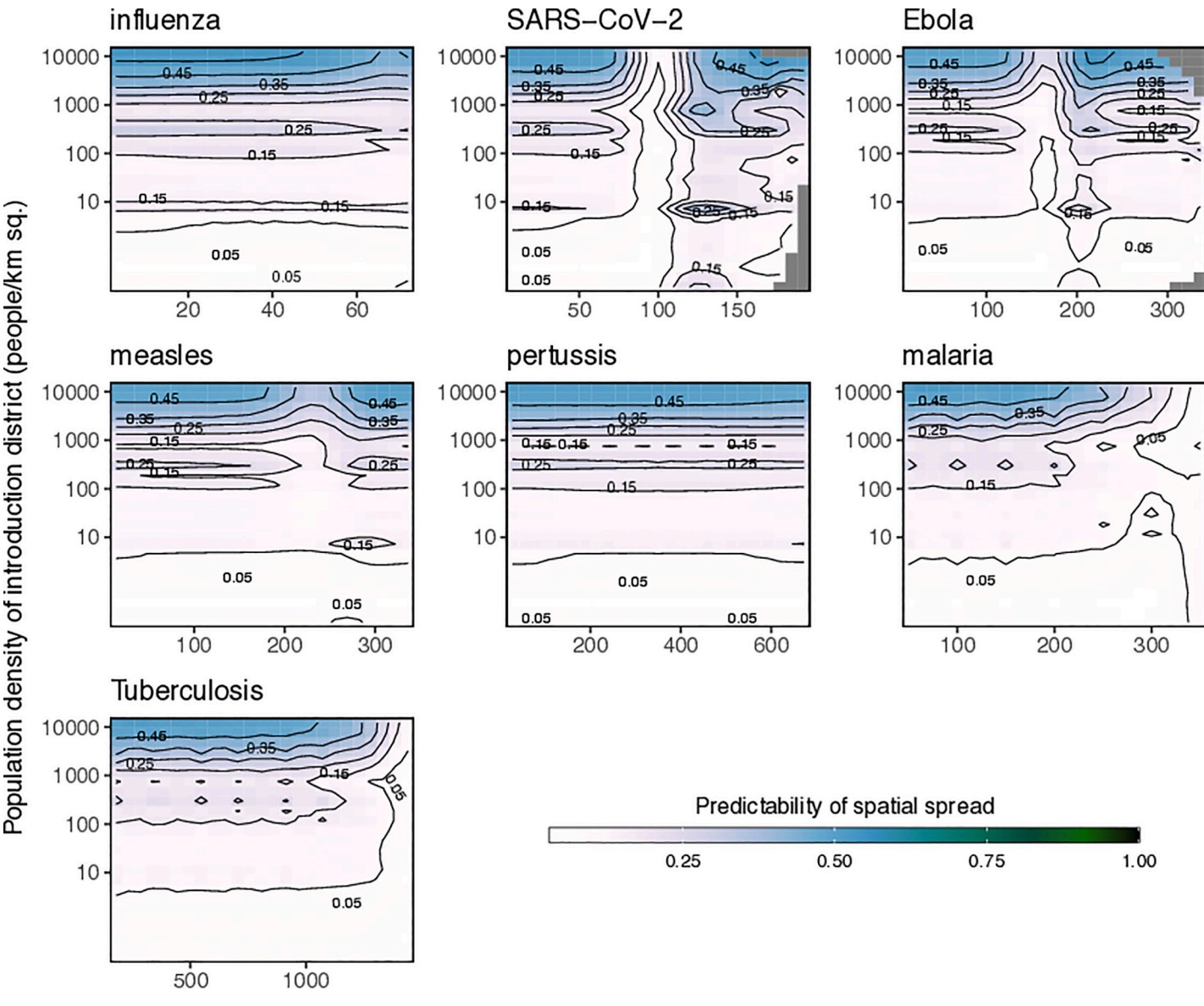

**Fig 7. The change in spatial predictability for 6 pathogens plotted over time for outbreaks introduced to districts with a range of population densities.** The results from simulations of outbreaks for 7 example pathogens (influenza, SARS-CoV-2, measles, Ebola, pertussis, *P. falciparum* malaria, and Tuberculosis). For each pathogen, outbreaks were introduced into each of the 105 districts in Namibia and the change in spatial predictability for successive generations was calculated. The heatmaps and contour lines show values of spatial predictability ($\phi$) as they change over successive generations (x-axis) and the population density of the introduction district (y-axis). The number of days since the introduction is indicated on the top-axis. Annotations in the lower right indicate the pathogen, basic reproduction number ($R_0$), generation time in days ($\gamma$), and proportion of the population that is susceptible (*s*) used in spatial simulations. Transmission parameters used in simulations were drawn from the literature and shown in S1 Table.

find that the spatial force of infection for pathogens with longer generation times is less influenced by the distribution of shorter trips because the duration of shorter trips comprises a much smaller proportion of the total infectious period, allowing for less infected-person time in the visited location. By comparison, longer duration trips contribute more towards the spatial force of infection for a long generation time pathogen because the duration of these trips has greater overlap with the infectious period, translating into more infectious days spent

travelling all else being equal. This proportional difference effectively dilutes the force of infection for longer generation time pathogens in discrete-time models, because individuals are assumed to be infectious for the entire timestep. Spatial transmission in this manner is most strongly driven by trips with a duration approximately equal to or greater than the generation time of the pathogen[30]. Therefore, in our disease simulations informed by the Namibian travel data, we see intrinsic spatial bias in the transmission process, where spatial spread of pathogens with shorter or longer generation times are potentially driven by vastly different patterns of spatial connectivity with fundamental effects on the overall spatial predictability of disease spread. For example, the spatial force of infection for a short generation time pathogen like influenza will be primarily driven by the short-trip duration subnetworks which have higher heterogeneity and clustering around high-density urban areas. By comparison, spatial transmission of pathogens with longer generation times (e.g. pertussis or malaria) will be primarily driven by the long-trip duration subnetworks that have more homogenous spatial structure and lower overall connectivity.

Although we mainly investigated human mobility as the primary driver of spatial dynamics, the generalizability of our results to other disease systems also depends on spatial patterns in demographic and epidemiological factors specific to the pathogen [65]. For example, transmission of childhood diseases, such as measles and rubella, depends on the rate at which children travel and the demographic structure in each location. Heterogeneity in demographics among locations could change the patterns of spatial spread or introduce additional uncertainty. Population-level susceptibility also plays a significant role in the timing of spatial spread and is particularly important for pathogens with low population-level immunity [66,67]. Although we include a simple susceptibility parameter in our models of spatial spread, spatial heterogeneity in population susceptibility would also impact the locations that are at highest risk of spatial spread, as seen in instances where accumulation of susceptible individuals occurs due to vaccine refusal [68] or spatially heterogeneous vaccination coverage [69]. Further, we used a discrete-time model with Susceptible-Infected-Recovered disease dynamics which assumes individuals are uniformly infectious for the entirety of a timestep. However, it is less clear how the interdependence between trip duration and pathogen life history plays out when considering the precise timing of a trip in relation to different stages of disease (e.g. exposed, infectious, recovered) and whether infection has direct impact on the ability to travel. Additional work should be done to explore how the observed distribution of trip duration and uncertainty around latent period and infectious period may cause different dispersal dynamics depending on pathogen life history.

Methods for including trip duration have been proposed that include both theoretical [47,49] and data-driven [51] approaches. These methods offer a representation of spatial disease transmission that incorporates additional nuances of human travel decisions, but the resulting patterns of spatial spread may ultimately tend toward more unpredictable behavior (a persistent challenge when predicting the behavior of nonlinear systems in ecology and epidemiology [19–22]). Approaches which account for uncertainty in human mobility and resulting disease dynamics therefore become especially important in order to make robust inferences about spatial transmission. For example, Kahn et al. [70] showed that pathogens with longer incubation periods have patterns of spatial spread that are less predictable, which has important implications for vaccination campaigns. While the authors did not explicitly look at trip duration and network topology as we do here, these results lend support to our findings that the biological properties of pathogens that determine their speed of spatial propagation interact with the travel behavior of humans in a pathogen-specific manner and suggests that this general relationship may be more widely applicable. However, the extent to which our results here fully generalize to disparate geographic settings is unclear due to the exceptionally

low population density across much of Namibia with highly clustered populations in a few urban areas. This unique population distribution causes high variability in the size of administrative units, which are reflected in the physical placement of the network nodes in our analyses. The shifts in network topology that we observe here may be quite extreme on account of the unique population distribution in Namibia and could potentially exhibit different patterns in a country with a larger or more evenly distributed population. Future research will therefore benefit from analyzing mobility data from different contexts to establish the best predictors of trip duration that are generalizable and develop a simple data-driven mechanism, which includes these additional sources of heterogeneity, that can be easily incorporated into spatial transmission models.

## Methods

### Ethics statement

The data used in this study was approved by the University of Southampton ERGO committee (ERGO ID 23647). No individuals were enrolled to participate because the data have been aggregated and de-identified.

### Geographic data

In addition to the CDR data, we used open source data to provide the static population sizes of each district ($N_i$) and the distances between all district centroids ($d_{ij}$; Fig 1). District population sizes were calculated by summing 2010 WorldPop Project (www.worldpop.org) estimates of the total number of people per 100m grid cell within each district (administrative level 2) in Namibia. District-level shapefiles were acquired from DIVA-GIS (www.diva-gis.org) and distances between districts were calculated as the Euclidean distance between district centroids.

### Definition of duration-restricted subnetworks

We performed initial analyses of travel network topology by comparing relative edge weights for travel networks based on three broad intervals of trip duration. Relative edge weights were defined as $w'_{ij} = [(w_{ij}|t_1, t_2)/\max(w_{ij})] \times 100$ such that each sub-network contained only trips where duration of travel ($t$) was within the interval [$t_1$, $t_2$] and is scaled by the overall maximum edge weight observed in the full travel network $\max(w_{ij})$. The broad intervals for this initial analysis were defined as 1–3, 7–14, and 30–60 days in duration (see Fig 2). To further explore how network topology changes on account of trip duration, we subset our data into 20 duration-restricted subnetworks reflecting intervals of trip duration that are analogous to the generation times of several infectious pathogens (e.g. 2–3 days for influenza, 14–21 days for Ebola) with duration intervals continuing up to 12 months (see Table 1).

### Measures of network centrality

To characterize the overall topological structure of each duration-restricted subnetwork in Table 1, we used two centrality measures: node strength (i.e. weighted degree per node) and node closeness (i.e. the cumulative weighted distance separating node $i$ from all other nodes) [71]. Here, nodes represent geographic locations (districts). Specifically, we calculated node strength as the out-strength of node i as $r_i = \sum_{j=1}^{n} a_{ij} w_{ij}$, where n is the total number of nodes, $a_{ij}$ is the adjacency matrix, and $w_{ij}$ is a matrix of weights given by the total number of trips between i→j nodes [62]. We calculated node closeness according to Opsahl et al. [71] using the 'tnet' R package [72], which is defined as $c_i = 1/\sum_{j=1}^{n} d(i, j|w_{ij}, \alpha)$, with tuning parameter

$\alpha$ = 0.5 so that both edge weight and the number of intermediate nodes are used to calculate the shortest path between two nodes. The statistical distributions of node strength $r_i$ and node closeness $c_i$ provide information on the level of spatial clustering in the travel network. Where, a heterogeneous network is characterized by network centrality measures with skewed and/or long-tailed distributions compared to a more homogeneous network which is characterized by centrality measures that have shorter-tailed distributions.

## Measuring the level of heterogeneity in travel networks

To further investigate structural changes in network typology across networks defined by different intervals of trip duration, we condensed the statistical distributions of node strength $r$ and node closeness $c$ into two singular metrics of network heterogeneity to facilitate comparison. A mathematical definition of these summary metrics of network heterogeneity has been previously described in Barrat et al. [73] where the first and second moment of the centrality measure's distribution is concisely translated into a single heterogeneity parameter, which we have dubbed $\eta$. Following Barrat et al. [73] we defined the heterogeneity parameter $\eta$ for the distributions of both node strength and closeness, denoted as $\eta_{\text{strength}}$ and $\eta_{\text{close}}$ respectively. For example, network heterogeneity as measured by node strength $r$ is:

$$\eta_{\text{strength}} = \frac{\langle r^2 \rangle - \langle r \rangle^2}{\langle r \rangle} \propto \frac{\langle r^2 \rangle}{\langle r \rangle}.$$

Where, the denominator here represents the expected value of node strength across all $i$ nodes $\langle r \rangle = \int_0^\infty r P(r) dr$ and the numerator represents the variance $\sigma^2 = \langle r^2 \rangle - \langle r \rangle^2$, where the second moment is $\langle r^2 \rangle = \int_1^\infty r^2 P(r) dr$. In this case of node strength, we estimated the statistical distribution of $r$ for all nodes in the given travel network using a kernel density estimator with an equally-weighted mixture of Gaussian kernels [74] as the probability density function $P(r)$. Defined as such, $\eta$ can be considered analogous to the normalized variance in the distribution of the centrality measure. Heterogeneity $\eta_{\text{strength}}$ and $\eta_{\text{close}}$ are thus heuristic measures that indicate the magnitude of fluctuations in node strength and closeness relative to the mean, such that a highly heterogenous network will have large variance compared to the mean (i.e. $\eta_{\text{strength}} > \langle r \rangle$) in comparison with a homogenous one (i.e. $\eta_{\text{strength}} \approx \langle r \rangle$), as illustrated in S5 Fig [73].

## Estimating shifts in travel network structure

To identify potential thresholds of trip duration at which travel network structure shifts from a heterogeneous to homogenous structure, we performed a multivariate changepoint analysis [75] based on the joint distribution of the heterogeneity metrics $\eta_r$ and $\eta_c$. We calculated these metrics for networks comprised of trips falling within each of the trip duration intervals in Table 1 and then estimated the changepoint network heterogeneity using the Energy Divisive algorithm in the 'ecp' R package [76] which estimates both the number and position of changepoints in a multivariate space. The algorithm estimates the $k$ changepoints hierarchically by sequentially segmenting observations into groups and iteratively maximizing the divergence of zero-, first-, and second-moment measures among the $k+1$ groups so that they are mutually independent and identically distributed [77]. We estimated the changepoints for a minimum group size of 2 with a significance level of 0.05 and 10,000 permutations. Based on the estimated changepoints in ($\eta_r$, $\eta_c$) space, we inferred the thresholds of trip duration at which the travel network shifts significantly from heterogeneous to homogeneous.

### Identifying drivers of travel network structure

To explore potential drivers of this shift, we examined the distribution of raw trip counts emanating from each origin and its dependence on several variables describing characteristics of the origin and destination: distance, total population size, population density (people/km$^2$), and total administrative area (km$^2$). Overall, we tested 13 candidate log-linear models with relevant combinations of these variables and quantified model fit using the Akaike Information Criterion (AIC = $2k-2\ln(L)$, where $k$ is the number of parameters in the model and $L$ is the estimated likelihood function). The best and most parsimonious model contained two covariates: distance between origin and destination $d_{ij}$, and population size of the destination $N_j$, so we fit this log-linear regression model to trip counts for each origin $i$ and trip duration interval $[t_1, t_2]$ in Table 1.

$$\log(y_i|t_1, t_2) = \alpha_i + \sum_j (\beta_i d_{ij} + \gamma_i N_j) + \epsilon_i$$

Here, the linear predictor is fitted to the duration-restricted trip counts ($y_i|t_1, t_2$) using a log link function with origin-level intercept $\alpha_i$, coefficients $\beta_i$ and $\gamma_i$, and error term $\epsilon_i$. The models were fit to each origin and duration interval using Maximum Likelihood Estimation. We then compared the effect of the two covariates on duration-restricted travel volume by calculating the effect size (fitted slope) of each origin-specific model and then plotted them against each trip duration interval.

### The impact of trip duration and network structure on spatial disease dynamics

We used a stochastic TSIR model [78,79] of transmission dynamics to estimate the probability of disease importation and to assess the predictability of spatial disease spread under different network structures. We assumed a single introduction of one infected individual in origin location $i$ where the overall proportion of the population susceptible is given by $\epsilon$. Local growth of infected individuals is given by the force of infection:

$$\lambda_{j,t+1} = \frac{\epsilon \beta S_{jt} (I_{jt} + m_{jt})^\alpha}{N_j}$$

and the changes in each disease state in each of the $j$ locations are given by the following difference equations:

$$S_{j,t+1} = S_{jt} - \lambda_{jt}$$
$$I_{j,t+1} = I_{jt} + \lambda_{jt} - \gamma I_{jt}$$
$$R_{j,t+1} = \gamma I_{jt}$$

Parameters for the transmission rate $\beta$ and recovery rate $\gamma$ were pathogen specific with parameters values defined by $R_0 = \beta/\gamma$ under the simplifying assumption that pathogens are infectious for the entirety of the generation time. Spatial transmission is driven by $m_{jt}$ which is the total number of migrant infected individuals that arrive at destination $j$ from all other locations.

$$m_{jt} = \sum_{i \neq j} I_{it} \cdot (\pi_{ij}|g)$$

The conditional term ($\pi_{ij}|g$) is the fitted value of connectivity $\pi_{ij}$ given the weighted number

of trips for generation time $g$ of the pathogen ($y_{ij}|g$). Connectivity values in $\pi_{ij}$ are the estimated probabilities that a trip will go to each of the $j$ destinations when it departs origin $i$, where each row in the matrix $\pi_{ij}$ sums to 1. Connectivity is conditioned on pathogen generation time via the weighted trip count ($y_{ij}|g$), which decays based on the proportion of the generation time covered by trip duration $t$:

$$(y_{ij}|g) = \sum_{t=1}^{T=g} y_{ijt} e^{-g/t}$$

The total effective trip count is then given by the sum of all weighted duration-restricted subnetworks up to a maximum duration that is equal to generation time $g$. Connectivity was then fitted to the total effective trip count using a gravity model with the following likelihood function:

$$(y_{ij}|g) \sim \text{Pois}(\pi_{ij}, \, N_i)$$

$$\pi_{ij} \propto \theta \left( \frac{N_i^{\omega_1} N_j^{\omega_2}}{d_{ij}^{\gamma}} \right).$$

Where, $\theta$ is the proportionality constant, $\omega_1$ and $\omega_2$ are exponential parameters that control the attractive force between the origin $N_i$ and destination $N_j$ population sizes, and $\gamma$ scales the distance penalty. Gravity model parameters were estimated using Bayesian inference and Markov Chain Monte Carlo (MCMC) sampling with the 'mobility' R package [80].

To calculate the general distribution of spatial predictability, we calculated the probability of importation from the origin $i$ into each of the $j$ destinations at time $t$ using the TSIR model defined above:

$$p_{ijt} = 1 - e^{-\epsilon\beta(S_{jt}/N_j)I_{it}\cdot\pi_{ij}|g}.$$

We then calculated the probability of disease importation for all unique combinations of $R_0$ from 1 to 30 and generation times from 1 to 180 days. The probability $p_{ijt}$ was calculated upon each generation within the simulations giving a timeseries of importation probability for each location. We calculated the uncertainty in importation to all other potential destinations as the Shannon entropy of the vector $p_i$ as $H(p_i) = -\sum_j p_{ij}\log_2(p_{ij})$. To more easily compare values of importation uncertainty across all scenarios of $R_0$ and generation time, we defined the overall predictability of spatial spread from the index location $i$ as $\phi_i$:

$$\phi_i = 1 - \left( \frac{H(p_i)}{H_{\max}(p_i)} \right).$$

Here the $H_{\max}(p_i)$ function represents the principle of maximum entropy [81] for a system with an equivalent number of locations n: $p_{i\cdot} = 1/n$ for all destinations $j \in \{1, \cdots, n\}$. This formulation gives a value of 0 for the least predictable scenario where probability of importation to all locations is equal (i.e. maximum entropy) and a value of 1 for a perfectly predictable scenario (a single location with importation probability of 1).

To incorporate all sources of uncertainty in our estimates of the predictability of spatial transmission, we performed 1000 replicates of each TSIR model. Upon each iteration, $R_0$ and generation time were allowed to vary within a plausible range for each pathogen (see S1 Table). Spatial connectivity estimated by the gravity model also varied according to the posterior distribution of estimated model parameters (i.e. $\theta$, $\omega_1$, $\omega_2$, and $\gamma$), and transmission events within the TSIR simulation occur with Binomial error structure. Analyses and simulations

were performed in the R programming language [82] using the 'mobility' [80] and 'hmob'[83] R packages.

## Supporting information

**S1 Fig.** The number of observed routes (A) and the number of observed trips (B) for each of the 20 duration-restricted travel networks.
(PDF)

**S2 Fig.** Distributions of node strength (A) and node closeness (B) for each of the 20 duration-restricted travel networks.
(PDF)

**S3 Fig.** Top: changes in the average expected value of for distributions of node strength $\langle r \rangle$ (A) and node closeness $\langle c \rangle$ (B) across all 20 of the duration-restricted travel networks. Bottom: changes in the variance of distribution of node strength $\langle r2 \rangle$ (C) and node closeness $\langle c2 \rangle$ (D) across all 20 of the duration-restricted travel networks.
(PDF)

**S4 Fig.** Changes in node strength (A) and node closeness (B) for each duration-restricted travel network according to population density of the origin district. Points show the observed values of node strength and node closeness dependent upon the log population density of each origin district. Trend lines indicate the changes in both networks measures across all values of population density. The color key to the right indicates the duration interval corresponding to each travel network.
(PDF)

**S5 Fig.** A) The relationship between the network heterogeneity metric based on node strength ($\eta_r$) and the average expected value of the observed distribution of node strength $\langle r \rangle$. B) The relationship between the network heterogeneity metric based on node closeness ($\eta_c$) and the average expected value of the observed distribution of node closeness $\langle c \rangle$. Dashed line indicates the x = y line and each point is colored according to duration-restricted travel network.
(PDF)

**S6 Fig. Change in goodness of fit across 20 duration-restrict gravity models.** Figure A shows the R-squared value of each model and figure B shows the Mean Absolute Percent Error (MAPE).
(PDF)

**S7 Fig. Changes in the gravity model parameters for each randomly down-sampled travel networks.** Down-sampled travel networks are the full travel network where the number of observed routes and number of observed trips is sampled randomly according to Table 1 in the main text. Figures A and C show changes in the distance parameter γ and population destination size parameter $\omega_2$ respectively. Figure B shows the effective penalty of distance ($d_{ij}^{\gamma}$) on estimated connectivity for each down-sampled network. Figure D shows the change in the attractive force of destination districts ($N_j^{\omega_2}$) based on the population size of the destination for each of the down-sampled networks.
(PDF)

**S8 Fig. Connectivity values fitted by gravity models for all randomly down-sampled sub-models are compared to connectivity of the full model (all data).** Down-sampled travel networks are the full travel network where the number of observed routes and number of observed trips is sampled randomly according to Table 1 in the main text. Red dashed line

indicates the x = y relationship.
(PDF)

**S9 Fig. Model fitting results from 13 candidate log-linear models used to estimate trip counts among origin and destination locations within each of the 20 trip duration intervals.** Covariates in the models include: distance between origin and destination, total population size (orig_pop, dest_pop), population density measured as people/km2 (orig_popdens, dest_popdens), and total administrative area (km$^2$; orig_area, dest_area). Model fit was assessed with the Akaike Infor- mation Criterion (AIC = 2k − 2ln(L), where k is the number of parameters in the model and L is the estimated likelihood function). The circles represent the mean AIC for models across all origin location and all duration intervals with 50% and 95% confidence intervals shown as thick and thin lines respectively. The lowest AIC value of any model is shown by the dashed red line (AIC = 371) and the selected model (count $\sim$ distance + dest_pop) is indicated in light blue.
(PDF)

**S10 Fig. The change in spatial predictability over subsequent generations for 7 pathogens.** Example pathogens are indicated by colored lines: influenza, SARS-CoV-2, measles, Ebola, pertussis, P. falciparum malaria, and tuberculosis (TB). Colored lines represent the mean value of spatial predictability over 1000 stochastic TSIR simulations with shaded regions showing the 95% confidence interval for the mean. In these simulations, the capital district of Namibia, Windhoek East, was used as the introduction district.
(PDF)

**S11 Fig. Change in spatial predictability depending on the number of destination locations that are predicted to have disease importation.** The left figure shows the theoretical change in spatial predictability for a travel network with one infected origin location and 104 potential destinations (n = 105). The red circle indicates the maximum value of spatial predictability ($\varphi$ = 0.56) based on spatial spread of several pathogens on the Namibia travel network. The figure on the right shows the change in spatial predictability for travel networks of up to n = 1000 locations. The travel network size is indicated by the color bar and the network with size equivalent to the Namibia travel network (n = 105) is highlighted in red.
(PDF)

**S1 Table. Transmission parameters for each of the six pathogens used in simulations of spatial spread.** Where, $R_0$ is the basic reproduction number, $g$ is the generation time in days, $s$ is the proportion of the population that is susceptible, and $1/\gamma$ is the infectious period in days.
(PDF)

## Author Contributions

**Conceptualization:** John R. Giles, Amy Wesolowski.

**Data curation:** Andrew J. Tatem, Amy Wesolowski.

**Formal analysis:** John R. Giles.

**Funding acquisition:** Andrew J. Tatem, Amy Wesolowski.

**Investigation:** John R. Giles.

**Methodology:** John R. Giles.

**Project administration:** Amy Wesolowski.

**Software:** John R. Giles.

**Supervision:** Amy Wesolowski.

**Visualization:** John R. Giles.

**Writing – original draft:** John R. Giles.

**Writing – review & editing:** John R. Giles, Derek A. T. Cummings, Bryan T. Grenfell, Andrew J. Tatem, Elisabeth zu Erbach-Schoenberg, CJE Metcalf, Amy Wesolowski.

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
