## [Decision Letter · Decision Letter 0]

2 Dec 2020

Dear Dr Giles,

Thank you very much for submitting your manuscript "Trip duration drives shift in travel network structure with implications for the predictability of spatial disease spread" for consideration at PLOS Computational Biology.

As with all papers reviewed by the journal, your manuscript was reviewed by members of the editorial board and by several independent reviewers. In light of the reviews (below this email), we would like to invite the resubmission of a significantly-revised version that takes into account the reviewers' comments.

We cannot make any decision about publication until we have seen the revised manuscript and your response to the reviewers' comments. Your revised manuscript is also likely to be sent to reviewers for further evaluation.

Sincerely,

Alex Perkins

Associate Editor

PLOS Computational Biology

Virginia Pitzer

Deputy Editor

PLOS Computational Biology

Reviewer's Responses to Questions

**Comments to the Authors:**

Reviewer #1: This work provides an engaging analysis of the manner in which networks of human trips change based on the duration of the trip. It then develops potential implications of the more diffuse pattern of longer-duration travel for predicting the spread of an infectious disease from a point of introduction within the area. Although I have few criticisms of the network analysis, I have a hard time accepting the implications presented without further explanation about the assumed low-level details of how transmission occurs within the contact network.

The authors filter the full network of district-to-district trips extracted from CDR by trip duration, generating 20 duration-restricted subnetworks. In the simulation of disease spread, the authors consider pathogens with a range of reproduction numbers and generation times. For coupling populations, they use a gravity model that is fit to the duration-restricted subnetwork with a duration matched to the generation time of the pathogen. I don't understand why such a restricted contact model is appropriate, especially since the longer duration subnetworks comprise a much smaller volume of the total trips in the data sets. It would seem to me that trips of all duration are potentially relevant to the spread of any of the pathogens considered, and trips with a duration equal to the generation time of the pathogen may be relatively unimportant if they are sufficiently rare.

The rationale of the authors for using the duration-restricted networks seems to be primarily in the second paragraph of the discussion. The authors cite Keeling and Rohani (2002, Ecology Letters) as a theoretical precedent but I don't see a direction connection between the results in that paper, which analyze the dynamics around an endemic equilibrium, and the scenario modeled by the authors of this manuscript, which is movement away from an unstable disease-free equilibrium. Nevertheless, the authors write "[Keeling and Rohani] show that when the trip duration is shorter than the infectious period of the pathogen, interdependence between pathogen transmission parameters and coupling among locations increases." The authors may be referring in part to figure 2b of that paper, which shows that the coupling parameter increases with the return rate of the trip. That result seem to support my contention that the authors should not neglect trips that are of shorter duration than the generation time in their simulations.

Keeling and Rohani (2002) also find that the proportion of time in a non-resident population is important for the extent of coupling. But this proportion depends not only on the duration of the trip (determined by the return rate tau in their model), but also on the frequency of the trip (determined by rho in their model). I point this detail out to put my comment in the previous paragraph about the rarity of long-duration trips making them potentially less important in the context of this theoretical work. Going back to the above quoted point in the manuscript, the authors next write "Accordingly, the spatial force of infection for pathogens with longer generation times is less skewed towards the distribution of shorter trips because the duration of shorter trips comprises a much smaller proportion of the total infectious period, allowing fewer opportunities for onward transmission." It seems from this sentence that I am not understanding exactly which results of Keeling and Rohani the authors have in mind, so I would recommend the authors being more explicit here. The authors are perhaps proposing some new theories in this and the following sentences based on the extent of overlap between trip duration and infectious period, which could very well be correct, but I find it difficult to understand them as written.

Minor comments:

1. The figures are generally well-constructed, but some figures (1, S3, S5) don't seem to have any references to them in the text. I just mention this in case it was an oversight of the authors.

2. The y-axis on figure S4B would be more informative if it read "Node closeness".

3. The authors write that "pathogens with longer generation times may see more diffusive patterns of spread that are not dominated by urban travel." I wonder whether this apparent diffuseness is an artifact of the fact that there seems to be only a single urban center in the network. Perhaps if the data were available for the entirety of Southern Africa, one might see that longer duration trips were more affected by distance and population size. If the author see some merit to these ideas, I would suggest they caution against generalizing the results about the change in network structure with trip duration beyond the study area.

4. Methods, Measures of network centrality, Is the a_ij in the equation for r_i a typo?

5. Methods, Impact of trip duration and network structure on spatial disease dynamics, should the equation for I_it = (beta s)^(t - 1) I would expect the susceptibility to affect growth in each generation.

Reviewer #2: Please see attached document.

Reviewer #3: Comments:

The study main’s objective is to investigate the effects of human mobility on infectious diseases spread dynamics, with particular attention to trips duration and spatial heterogeneity. While this is a very interesting topic, and I appreciated the approach to subdivide the network according to trip duration, the manuscript is presented in a quite confusing flow and the results present little novelty.

The analysis on the trip duration defined networks is interesting from a network analysis perspective, although the choice of using 20 different sub-networks is unclear to me (se also comments below). It is also unclear why they imposed 3 classes of network, instead of estimating an optimal number of classes as well using the changepoint algorithm. This could be later connected with the epidemiology of diseases, providing a more fluid framework.

In my opinion, the predictability analysis needs to be reviewed. The variability embedded in most diseases’ transmission dynamics is hardly considered, which makes it really hard to draw strong conclusions. First, it is unclear if the diseases spread has been calculated once in a deterministic way (using the model described in the methods) or over a number of simulations. Furthermore, All the parameters included (R0 and the generation time) are considered as points estimates, rather than distributions. As showed by the currently ongoing pandemic, the variance of the new cases generated by a single case can make a big difference, as well as the variability in generation time distributions. Here this aspect is not considered. This results in the analysis returning quite predictable results. For this reason, I believe that using single trips duration networks might be quite limiting, and (connecting with my previous point) would be more useful to consider an ensemble of sub-networks which could account for the generation time variability.

The choice of the diseases in the last analysis is odd. First, I would not consider malaria as a candidate, given its complex dynamics heavily affected by other environmental, climatic and ecological factors. These might indeed obscure the mobility and connectivity aspects.

Also, it is not clear why disease with slower dynamics are not considered (e.g. TB or HIV), since the data permit longer generation time analyses.

The manuscript would benefit from using empirical data of a real disease, to confirm the authors findings, but I understand if these are not available.

Overall, the manuscript of a clear logic flow, it has been very challenging to understand all the analysis and their purpose in the study.

Minor comments

General

Even if it might not be specified by the journal guidelines, for next time please consider inserting line numbering and page numbers, and maybe use a 1.5 or 2 line spacing. These would quite help the reviewers.

Abstract

• 2nd sentence: very confusing, please rewrite.

• 3rd sentence: see above.

• 4th sentence: “spatial distribution”

• 8th line: “imbedded”?

• Final sentence: please consider shortening.

Introduction

This section needs a strong revision. Sentences are often confusing and too long, and there is a lack of punctuation. From the point of view of content, it might also benefit from a re-organisation with a clearer flow and a better introduction of specific concepts. At the end of this section, the novelties presented by this study are not clear.

1st paragraph

• 4th line: given that most of the COVID-19 pandemic happened in 2020 (so far), consider remove “2019” or add “started at the end of”.

• 4th sentence: reference needed.

• 5th sentence: isn’t this already stated at the beginning of the section? (1st sentence)

• 7th sentence: I would say that the challenges are caused by the complexity of the phenomenon we want to model, not because models “often exhibit complex dynamics “.

3rd paragraph

• 2nd sentence: the part after the comma is not very clear, and please introduce “return rate” meaning.

Results

General: many of the results are discussed in this section, and this creates an overlap with the discussion section. Please, consider presenting the results in a “cleaner” way, and then discuss them accordingly in the discussion section.

Figure 1 is never mentioned in the text, maybe it should go somewhere in the first Results paragraph.

2nd paragraph

• 1st sentence: “we assessed travel by trip duration” please clarify.

• 3rd sentence: the phrase “duration-restricted subnetwork” is not very clear, maybe something like “subnetworks defined by trip duration” or similar would be more intuitive for the reader.

• 4th sentence: weighted by what?

• 6th sentence: I am not sure of the meaning of this sentence, isn’t the network clustering captured by the global clustering coefficient? Also please make clear that you calculated the heterogeneity according to Barrat’s formula.

• 6th to 9th sentences: this section would better fit the discussion. Also, a bit confusing and not entirely sure how useful is to discuss nodes strength or closeness “values” rather than their distribution.

• 10th sentence: by looking at figure S4 (and figure 3) I cannot see what the authors stated here. In particular, for short trip networks the closeness distribution looks quite “normaloid”, rather than heavy tailed.

3rd paragraph

• 1st sentence: this would better fit the discussion. Also, please make clear where are the variances of strength and closeness showed.

• Last two sentences: this would better fit the discussion.

4th paragraph

• Figure 4: this figure is very confusing, really hard to understand the message it tries to convey (if it is the findings described in this paragraph).

5th paragraph

• 1st sentence: “the most commonly used spatial interaction model”, citation needed. Moreover, the logic connection between the sentence and the table is not clear.

• Figure 5/A: is the reason that there are more than 1 value of connectivity for the full network because of subsampling? Please, clarify.

• R-squared � R2

6th paragraph

1st, 2nd and 3rd sentence: discussion.

Discussion

1st paragraph

• 1st sentence: citation needed.

2nd paragraph

• 3rd sentence: unclear, please rewrite.

• Last sentence: this is quite trivial.

3rd paragraph

• 1st sentence: citation needed.

•

Methods

2nd paragraph: I am wondering which epidemiological difference it makes considering as separate networks built with 10 and 11 months (as a random example). Maybe longer period could be considered together (6-9 months, 9-12 months) to make the results easier to read, unless there is an epidemiological reason to keep them separate. This is also because no one of the considered disease has such long generation time, as well as the maximum assumed generation time is 30 days.

4th paragraph: a bit confused, please rewrite. What is the purpose to use a gaussian kernel to estimate a statistical distribution, why the empirical could not be used? Moreover, it is not clear if at the end of this calculation we have 1 n_strength and 1 n_close value per each network, or a distribution of values.

5th paragraph: I am not familiar with the specific algorithm used, but from what I read it looks like a clustering one. “Enforcing” is a quite odd wording choice.

Last section: this needs a better explanation, it is really hard to follow and to understand the justification to the authors’ choices. The connectivity parameter is calculated using a gravity model which is fitted to the observed data, correct? “Connectivity” is a vague term in this context, since in graph theory has other meanings (unless it has already been used with this precise formula, hence a reference would be necessary). Also, how does the transmission model accounts for the decline in susceptibility after the first generation?

Sup mat

Table S1: the reference for sars-cov-2 parameters is to a sars-cov-1 paper (2004). About the Ebola one, please considering to refer to more recent retrospective modelling studies rather than one published while the outbreak was still ongoing. Also, which influenza virus is considered here? The reference points to a book on disease modelling, quite vague.

**Have all data underlying the figures and results presented in the manuscript been provided?**

Reviewer #1: **No: **The authors provide a link to the third party that provided the CDR, but there is no explanation given for why the authors could not share some aggregated form of the data which is analyzed in this paper. If they cannot do that due to an agreement with the data provider or because of privacy concerns, they should state that but if the authors are able to make such data available, it would greatly enhance the reproducibility of the reported results.

Reviewer #2: Yes

Reviewer #3: **No: **

PLOS authors have the option to publish the peer review history of their article (what does this mean?). If published, this will include your full peer review and any attached files.

Reviewer #1: **Yes: **Eamon B. O'Dea

Reviewer #2: **Yes: **Greg Albery

Reviewer #3: No
---

## [Decision Letter · Decision Letter 1]

28 May 2021

Dear Dr Giles,

We are pleased to inform you that your manuscript 'Trip duration drives shift in travel network structure with implications for the predictability of spatial disease spread' has been provisionally accepted for publication in PLOS Computational Biology.

Best regards,

Alex Perkins

Associate Editor

PLOS Computational Biology

Virginia Pitzer

Deputy Editor-in-Chief

PLOS Computational Biology

Reviewer's Responses to Questions

**Comments to the Authors:**

Reviewer #1: The authors have addressed my main concern by no longer ignoring the effect of the short-duration trips. I would have preferred a clearer and more mechanistic derivation of the weighting scheme the authors use to mix trips of different duration. At any rate, the authors have clearly generated and described results with there own choice, and these results may prove useful to some readers in their own right. The authors have also taken care to address the other items I noted in my first review.

Minor comment:

line 322: Addition work  Additional work

Reviewer #2: The authors have done a great job of addressing my comments. I have two very minor recommendations, but I advocate for publication.

All the best,

Greg Albery

• Abstract: “models of travel are often used that assume” could be “models of travel often assume”

• Nowadays “embed” is a much more common spelling than “imbed”, and the use of the latter had me googling whether there was some specific meaning in this context that I might be missing. I recommend changing to “embed” to avoid this for other readers.

Reviewer #3: The authors did a good job addressing all my and other reviewers' comments, therefore the manuscript is now fit for publication in PCB.

Very minor comment: maybe add a sentence somewhere in the text (Res or Mat&Met) stating what a TSIR model is.

**Have the authors made all data and (if applicable) computational code underlying the findings in their manuscript fully available?**

Reviewer #1: **No: **Code is available on request, so its future availability is unclear.

Reviewer #2: Yes

Reviewer #3: **No: **

PLOS authors have the option to publish the peer review history of their article (what does this mean?). If published, this will include your full peer review and any attached files.

Reviewer #1: **Yes: **Eamon O'Dea

Reviewer #2: **Yes: **Greg Albery

Reviewer #3: No

---

## [Editor Report · Acceptance letter]

7 Jul 2021

PCOMPBIOL-D-20-01900R1 

Trip duration drives shift in travel network structure with implications for the predictability of spatial disease spread

Dear Dr Giles,

I am pleased to inform you that your manuscript has been formally accepted for publication in PLOS Computational Biology. Your manuscript is now with our production department and you will be notified of the publication date in due course.

With kind regards,

Kata Acsay
